# Lumbar Multifidus Muscle Morphology Changes in Patient with Different Degrees of Lumbar Disc Herniation: An Ultrasonographic Study

**DOI:** 10.3390/medicina57070699

**Published:** 2021-07-08

**Authors:** Neda Naghdi, Mohammad Ali Mohseni-Bandpei, Morteza Taghipour, Nahid Rahmani

**Affiliations:** 1Department of Health, Kinesiology & Applied Physiology, Concordia University, Montreal, QC H4B 1R6, Canada; nd.naghdi@gmail.com; 2Pediatric Neurorehabilitation Research Center, University of Social Welfare and Rehabilitation Sciences, Tehran 1985713834, Iran; nahrah2005@yahoo.com; 3Department of Physiotherapy, University of Social Welfare and Rehabilitation Sciences, Tehran 1985713834, Iran; morteza.taghipour.pt16@gmail.com

**Keywords:** lumbar disc herniation, lumbar multifidus muscles, ultrasonography

## Abstract

*Background and Objective*: Previous studies demonstrated that the prevalence of lumbar disc herniation (LDH) is relatively high. This investigation aimed to evaluate the size of lumbar multifidus (LM) muscle in patients with different degrees of LDH compared to healthy group, during rest and contraction, using ultrasonography. *Materials and Methods*: In this non-experimental, analytic, and case control study, ultrasound imaging was used to assess cross-sectional area (CSA) and thickness of the LM muscle in 15 healthy subjects and 60 patients with different stages of LDH (bulging group = 15, protrusion group = 15, extrusion group = 15, sequestration group = 15). Measurements were taken bilaterally at the L4–L5 level, during rest and contraction and results were compared between groups. *Results*: There was a significant difference between healthy subjects and the extrusion and sequestration groups during rest and contraction for LM muscle CSA and thickness (*p* = 0.001), as LM muscle CSA and thickness were significantly smaller in extrusion and sequestration patient groups compared to healthy subjects. LM atrophy was greater in patients with extrusion and sequestration groups than in patients with bulging and protrusion, both at rest and during contraction. Significant correlations were also observed between functional disability and intensity of pain with LM CSA and thickness measurements. *Conclusions*: Patients with extrusion and sequestration LDH had smaller LM muscle at rest and during contraction compared to healthy subjects. Larger LDH lesions were associated with decreased LM muscle size. Patient with more pain, disability, and extrusion and sequestration LDH had greater LM size changes. LM muscle size was not correlated with symptom duration. Further investigation with greater sample size is warranted.

## 1. Introduction

Low back pain (LBP) is one of the most common and disabling musculoskeletal disorder worldwide [1,2,3]. Previous studies have shown that LBP is responsible for significant costs to individuals and society [1,2,3,4]. Lumbar disc herniation (LDH) is a common spinal disorder, which can lead to LBP and radicular pain [3,5,6]. Although the life-time prevalence of LDH is 3–5%, it has been suggested that LDH is the cause of about 90% of radiculopathies [7,8]. According to the lesion type and shape, herniated discs are categorized as bulging, protrusion, extrusion, and sequestration [9]. Compression by a protruding disc on the dorsal and/or ventral rami of the nerve roots can cause LBP, sensory loss (paresthesia, numbness and tingling), radicular leg pain, muscle weakness, and restriction of trunk movement [10]. Dysfunction of the back muscles is very common in patients with LDH [10]. Paraspinal muscles are important to maintain normal function and stability of the lumbar spine [11]. Among the paraspinal muscles, the lumbar multifidus (LM) muscle has been of particular interest as an active stabilizer of the lumbar spine. This muscle provides segmental stabilization by maintaining a neutral intervertebral position during functional activities [12,13].

The LM muscle is the largest and most medial of the lumbar paraspinal muscles [14]. This muscle is innervated by the medial branch of the posterior root of the segmental nerve, and its innervation is thought to be unisegmental [13]. LDH leading to nerve root compression was associated with important paraspinal muscle degenerative changes, including significant LM muscle atrophy localized at the pathological level and symptomatic side (10). Such findings were reported in studies including patients with LHD [15,16], nerve root lesions [17], disc degeneration [18] nerve root avulsion [19], as well as radiculopathy or unilateral acute or chronic LBP [12,20,21,22]. Analysis of the paraspinal muscles is essential for determining the proper methods and intensity of exercise of the lumbar spine [15]. While exercise therapies have been evolved for LBP patients [1], recent emphasis is placed on specific exercises that aim to restore the stability in the lumbar spine [6]. It is believed that the mechanism for pain relief with this specific type of exercise is through enhanced stability of the lumbar spine segments, specifically targeting the LM muscle [6,12]. More research on patients with LBP needing lumbar stability exercise therapy is required to provide evidence of efficacy of the exercise therapy in this group of patients [22].

Among various methods, including computed tomography (CT) [20], electromyography (EMG) [2], magnetic resonance imaging (MRI) [23,24,25], and ultrasonography [26], ultrasound is reported to be an accessible, non-invasive, valid, and reliable imaging technique which is widely used to assess the muscle structure, function, and activity [27,28,29,30]. In addition, the main advantages of ultrasound are being radiation free, cost-effective, and well tolerated by patients, as well as appropriate for serial follow-up [31,32].

In general, previous studies have shown that patients with LDH have smaller trunk muscles (e.g., psoas major and LM muscles), as compared to healthy asymptomatic subjects [6,15]. However, it remains unclear whether the degree of muscle atrophy is associated with the LDH classification. To our knowledge, there is a lack of research on the evaluation of cross-sectional area (CSA) and thickness of the LM muscle among different stages of LDH separately and in comparison with healthy subjects. Therefore, the objective of this study was to evaluate the CSA and anteroposterior (AP) thickness of LM muscle in patients with different degrees of LDH and with concomitant unilateral radicular symptoms using ultrasound.

## 2. Materials and Methods

### 2.1. Study Design and Subjects

The present project was approved by the ethical committee at The University of Social Welfare and Rehabilitation Sciences Tehran, Iran. Sixty patients (32 females and 28 males) with different degrees of unilateral LDH participated in this study. Based on the MRI findings and according to the degree of LDH, the patients were diagnosed and divided into 4 groups: bulging, protrusion, extrusion, and sequestration with a total of 15 patients (8 females and 7 males) in each group. A prominence of annulus fibrosus external fibers extending beyond the edge of the vertebral body is defined as disc bulging [33]. Disc protrusion was used when the width of the protruded disc was lesser than the distance between the edge of the base, and disc extrusion was used when the width of the protruded disc was greater than the distance between the edge of the base [34,35]. A sequestrated disc is referred to as a loss of continuity of the disc material or displaced disc [35].

Patients were included if they had (1) posterolateral LDH at the L4–L5 level only, with ipsilateral radiculopathy or leg/pain symptoms, (2) symptoms duration ≥3 months, (3) no history of lumbar surgery, and were (4) aged between 20 and 60 years old. Exclusion criteria included subjects with LDH in other levels of lumbar or diagnosed with other spinal abnormalities (e.g., spinal stenosis or spondylolisthesis), any history of a sacroiliac joint dysfunction, lumbar fracture, spinal deformity or scoliosis, rheumatologic and neurologic disease, orthopedic device in the spinal column, pregnancy, metabolic diseases and malignancies or other major medical conditions and sensitivity to sonography gel.

Patients were recruited by spinal surgeon clinics in Tehran, Iran. All patients underwent a conventional neurologic examination and were diagnosed with LDH by a spinal surgeon. Level and degrees of LDH were recorded using MRI. According to inclusion/exclusion criteria, subjects were then assessed for being eligible at the time of their first visit in the spinal clinic and then eligible patients were selected and referred to research group.

A total of 15 healthy volunteers (8 females and 7 males) without LBP were chosen for comparison. The participants of healthy group were matched to the patient’s group for age, weight, height, and body mass index (BMI). In this study, right LM in the healthy group was considered as the measurement size for between group comparison. The left one was measured to check the normal distribution of data.

### 2.2. Procedure

Eligible subjects were asked to read and sign the consent form to participate in this investigation. Demographic data including age, duration of symptoms, gender, height, and weight were collected. Visual analogue scale (VAS) [36] and Oswestry Low back pain disability index (ODI) [37] were used to measure LBP intensity and related disability, respectively.

Ultrasound images were obtained using ES500 ultrasound machine (Ultrasonix-ES500, Canada) in B-mode with a 7.50 MHz, 70 mm curvilinear head transducer to measure LM muscle size [38]. All images were taken at rest and during contraction. To measure the LM muscle size, the lumbar curve was measured by an inclinometer in the standing position. Then, subjects were asked to lay in the prone position with 1 or 2 pillows under their pelvis to keep the standing lordosis [39]. To identify the LM muscle, the line connecting the iliac crests to the lumbar spine was considered as a landmark to detect L4/L5 spinous process. The spinous processes placement was then validated using ultrasound in sagittal view imaging in which the L4–L5 spinous process is located next to the sacrum. The ultrasound probe was perpendicularly placed at the L4/L5 vertebral body. Then, the CSA and AP thickness of LM muscle were captured bilaterally [40]. During contraction, subjects were asked to lift their contralateral arm about 2 cm off the table while elbows were flexed about 90° and shoulders abducted about 120° [22,41].

To measure CSA of the LM, thoracolumbar fascia and lamina were used to recognize the superior and inferior boundary of the LM, respectively. In addition, the LM is medially bordered by acoustic shadow of spinous process. The lateral boundary was provided using the fascia, which separates the LM from the lumbar erector spine muscle. The AP of LM thickness measurement was obtained by tracing the distance between the superior and inferior boundaries of the LM muscle CSA (Figure 1) [42]. Three repetitions were performed for each image and the average value was recorded and used in the analysis to reduce measurement error. Good to excellent within-day and between-day interrater reliability and within-day intrarater reliability from the same rater (intraclass correlation coefficients ranged from 0.70 to 0.91) were previously reported for CSA and AP thickness of the LM muscle measurements using ultrasound in patients with unilateral LDH, both at rest and contraction [42].

### 2.3. Data Analysis

Statistical analysis was performed using SPSS for Windows, version 19.00. Kolmogorov-Smirnov test was applied to assess the normal distribution of data and *p*-value > 0.05 was observed in all parameters. One-way ANOVA with post hoc Bonferroni tests was used to compare the CSA and AP thickness of LM between each patient groups with the healthy group, as well as between patient groups. The relation between LM muscle size and patients’ clinical outcome variables was measured using Pearson correlation coefficient (r). A *p* < 0.05 level of significant was set for the analysis.

## 3. Results

Demographic characteristics of subjects are presented in Table 1. The mean and the standard deviation of LM muscle CSA and AP thickness in patients and healthy group are presented in Table 2.

The results of One-way ANOVA with post hoc Bonferroni tests to compare LM muscle CSA and AP thickness between each patient group with healthy group are demonstrated in Table 3. Our findings revealed that although smaller LM muscle CSA and thickness were found in patient groups with LDH as compared to healthy control subjects, this difference was only significant in the extrusion and sequestration groups at both rest and contraction (*p*-value = 0.001). The *p*-value of 0.02 for AP thickness mean difference was observed in protrusion group as compared to healthy subjects during rest.

As shown in Table 3, extrusion and sequestration LDH lesions led to greater LM muscle measurements mean difference when compared to healthy subjects. Furthermore, when comparing LM size at rest and during contraction between the patient groups, LM CSA and thickness decreased, but the CSA difference was only significant between bulging group and extrusion group at rest (*p*-value = 0.002), bulging group and sequestration group at both rest and contraction (*p*-value = 0.001) and between protrusion group and sequestration group at contraction (*p*-value = 0.01). For the AP thickness mean difference, the measurements for both rest and contraction were significant (*p*-value = 0.001) between bulging group and extrusion and sequestration groups. In addition, this difference was significant between protrusion and extrusion and sequestration (*p*-value = 0.02 and 0.004 respectively).

The correlation coefficients between pain intensity and functional disability index with LM muscle CSA and AP thickness of all patients (e.g., all four groups combined) are reported in Table 4. Significant negative correlations were found between pain intensity and functional disability index LM muscle CSA and AP thickness in patients with LDH (*p* = 0.001).

Furthermore, no association was found between LM muscle CSA and AP thickness and duration of symptom of each group. (*r* = −0.12–0.571) and (*p* = 0.124–0.67).

## 4. Discussion

The purpose of the current study was to evaluate the CSA and thickness changes of the LM muscle in patients with different stages of LDH and healthy subjects. In addition, the correlations between CSA and thickness of LM muscle measurements and pain intensity, functional disability level and symptom duration were identified.

The results of the present study are in line with our defined hypothesis regarding LM muscle innervation; extrusion and sequestration of the LDH increase the amount of nerve root compression in the lumbosacral region, which subsequently lead to LM muscle atrophy. As mentioned in the result, the CSA and thickness changes of the LM were greater in the sequestration group than the bulging group as compared to the healthy group. Moreover, between-group comparisons revealed greater CSA and thickness atrophy of the LM in patients with disc extrusion and sequestration, both at rest and during contraction.

Previous imaging studies generally assessed the LM morphological changes in patients with LDH, and did not consider the classification of the LDH. In the present study, LM muscle was characterized in four stages of LDH, both at rest and during contraction.

Changes in the LM muscle were previously observed in patients with LDH. Zhao et al. [10] and Yoshihara et al. [43] conducted histological investigations and showed that types I and II fibers on the side of the LDH were significantly smaller than those on the normal side. In animal studies, Dulor et al. [44] found that denervation of the skeletal muscle led to rapid muscle atrophy and replacement of connective and fatty tissues. Hodges et al. [17] revealed that LM muscle CSA was reduced after experimental disc and nerve root injury, which may be due to disuse following reflex inhibitory mechanisms. LDH is a major cause of lumbar radiculopathy, and several mechanisms, including mechanical compression and biochemical injury induced by herniated nucleus pulposus, [15] were proposed. Hodges et al. [17] presumed that the disc injury is unlikely to have influenced the root directly, but biochemical changes after disc injury can occur, and this was confirmed by electrodiagnostic studies in animal study [45]. Degenerative changes of the LM muscle were also found in patients with LDH. Hyun et al. [15] conducted a prospective clinical study examining the CSA of the LM muscle in patients with unilateral lumbosacral radiculopathy, lumbosacral disc herniation, and in healthy volunteers. They demonstrated that LM muscle CSA was reduced in both patient groups as compared to the controls and that LM muscle atrophy occurred more frequently in patients with radiculopathy than in those with LDH only.

LBP is one of the main symptoms in LDH. It can be caused by nerve root irritation and muscle spasm. Alston et al. [46] measured the strength of the trunk flexors and extensors by means of a cable tensiometer and reported that patients with chronic LBP have generalized weakness of the trunk muscle. They suggested that this weakness was due to inactivity imposed by pain or fear of pain. In addition, inactivity, reduced workload, and bed rest after LBP may lead to LM atrophy [10]. In addition, Danneels et al. [47] suggested that pain is responsible for inhibition of the stabilizing muscles by a combination of reflex inhibition and changes in coordination of the trunk muscles.

Kader et al. [11] also found bilateral and multilevel muscle degeneration, even in patients with single nerve root irritation. They postulated that LM muscle atrophy was due to lumbar dorsal ramus syndrome, with referred leg pain induced by irritation to structures innervated by the dorsal ramus nerve. Alternatively, neural drive to the LM muscle may be reduced by an inhibitory process, such as reflex inhibition, involving afferent discharge from the mechanoreceptors in the disc. Reduced activity due to inhibition is likely the result in disuse-related muscle changes. The reasons for this study’s compatibility with the mentioned studies can be found in the similarity of age, gender of the participants, and evaluation of the herniation involvement level.

In contrast to the present study results, Battié and colleagues [24] have found a greater CSA of the LM on the same level of the herniated side. Altinkaya et al. [14] showed that the mean of CSA of the LM muscle was not significantly different between the affected and the normal side.

In addition, a study by Fortin et al. [25] aimed to assess LM structural alteration in patients with unilateral L4–L5 LDH. CSA and functional cross section area (FCSA) of the LM were obtained on MRI images from L3–S1 levels bilaterally. There were no significant difference of CSA and FCSA asymmetry of the LM at L4–L5 level and adjacent levels. In this study it has been expressed that while some studies believe that LM muscles are innervated by dorsal branch of the posterior root of the unisegmental nerve, there are several other studies arguing that the LM muscles innervation is polysegmental [48,49]. The reasons for the inconsistency of this study with the present study can be firstly the differences in imaging modalities (ultrasound versus MRI), secondly, the differences in sample size, and thirdly, the heterogeneity of the cases. Meanwhile, in the present study, each stage of herniation was examined as a separate group, while a mixture of patients with different degrees of LDH was considered in Fortin’s study [25].

Findings of the current study are also in disagreement with the finding of Farshad et al. [50], who showed that the asymmetry of the LM muscle does not correlate with the severity of nerve root compression in the lumbar spine. However severe asymmetry with substantial LM atrophy seems to be associated with the probability of an indication of surgical decompression. 

The LM muscle provides an important contribution for the control and stability of the spine [51,52]. LM atrophy is likely to compromise the ability of this muscle to control the spine, which may have long term consequences. Recent data suggest that the recurrence of LBP is more likely in people who do not undertake a specific exercise strategies to restore the activity of the LM [53]. Rehabilitation exercises have been recommended for patients with paraspinal muscle dysfunction [15,18,53]. An exercise program conducted following lumbar discectomy reportedly improved outcomes with respect to pain, disability, and functional recovery [54]. Choi et al. [55] proposed the positive effects of the postoperative early lumbar extension muscle-strengthening program on pain, duration of improvement, and strength of back muscles in patients after operation of a herniated lumbar disc. The results of the present study also strongly support the necessity of an exercise program in such patients.

We did not observe correlation between LM muscle size and the symptom duration of the LDH. Similarly, Fortin et al. [31] investigated the relationship between LM morphology (asymmetry and composition) and duration of symptom in patients with LDH and found no association between them. Moreover, in another study by Chen et al. [56], morphological measurements were not correlated with symptom duration in patients with spinal stenosis of L4–L5.

The relatively small number of patients in each group and the inability to keep the assessor blind were among the limitations of the present study. In addition, symptom duration and the severity of symptoms as possible confounding factors of muscle atrophy were not considered in this study. Matching the symptom duration and the severity of symptoms in different groups of examination, evaluation and comparison of LM muscle morphology in a larger sample size in patients with different degrees of LDH; comparison of this method with other diagnostic methods such as electromyography and MRI; evaluation of LM muscle size in deferent levels of lumbar spine; conducting study by comparing different age groups with gender segregation, designing the therapeutic protocols and monitoring their effects, and using mean of right and left LM size measurement in the healthy group instead of only the right side are some suggestions for future studies. The most important clinical application of the present study is to achieve a deeper understanding of the LM muscle morphology changes (as stabilizing muscles of the spine) in people with various degrees of LDH to consider the possible preventive measures from the LDH.

## 5. Conclusions

A significant decrease of LM muscle size both at rest and during contraction was observed in patient with LDH extrusion and sequestration as compared to healthy asymptomatic subjects. A significant correlation was also observed between functional disability, LBP intensity, and muscle size. Further investigation with a greater sample size is suggested to confirm the results of the present study.

## Figures and Tables

**Figure 1 medicina-57-00699-f001:**
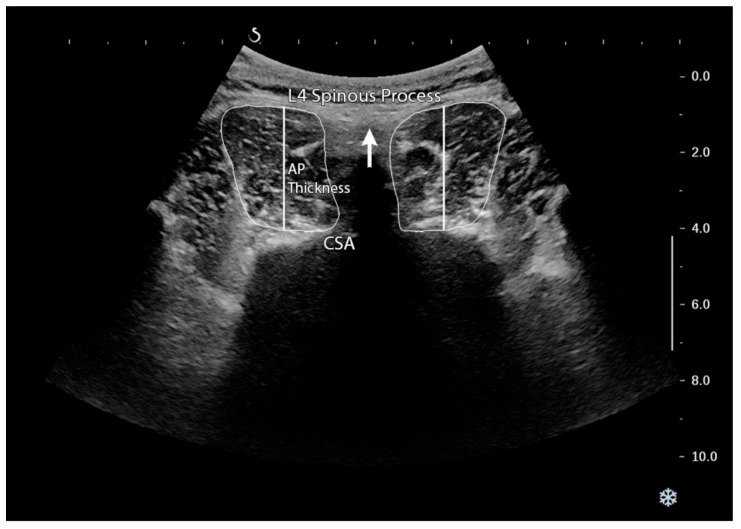
Cross-sectional area (CSA) and anteroposterior (AP) thickness measurements at the L4–L5 vertebral level.

**Table 1 medicina-57-00699-t001:** Demographic and clinical characteristics of subjects (Mean ± SD).

	Group	Healthy Subject (*n* = 15)	LDH Group			
Variable			Bulging (*n* = 15)	Protrusion (*n* = 15)	Extrusion (*n* = 15)	Sequestration (*n* = 15)
Age (years)		38.13 (14.38)	39.46 (11.01)	39.8 (9.39)	47.26 (7.08)	44.13 (8.35)
Weight (kg)		69.53 (8.69)	73.86 (9.39)	73.66 (12.79)	67.53 (10.25)	72.66 (9.72)
Height (cm)		169.80 (4.84)	169.40 (8.28)	175.33 (6.54)	169.06 (8.77)	172.6 (8.35)
BMI (kg/m^2^)		24.06 (2.21)	25.84 (3.88)	23.88 (3.54)	23.62 (3.21)	24.33 (2.64)
VAS (point)		-	2.60 (1.12)	4.33 (1.23)	6.00 (1.06)	8.33 (1.11)
ODI (point)		-	24.26 (3.10)	46.66 (3.51)	69.60 (3.13)	93.60 (3.64)
Symptom’s duration (m)		-	10.53 (1.77)	18.4 (2.29)	28.93 (2.93)	31.86 (2.8)

Abbreviations. SD: Standard deviation; m: Months; LDH: Lumbar disc herniation; BMI: Body mass index; VAS: Visual analogue scale; ODI: Oswestry disability index.

**Table 2 medicina-57-00699-t002:** LM muscle CSA and thickness measurements in LDH patients and healthy groups (at rest/contraction).

Muscle	State	Side	Groups	Mean (SD)	95% CI
CSA (cm^2^)	Rest	Right	Healthy	4.44 (0.73)	4.03–4.85
		Left		4.09 (0.67)	3.72–4.46
	Contraction	Right		5.16 (0.66)	4.79–5.53
		Left		5.24 (0.71)	4.84–5.63
AP (LM) Thickness (mm)	Rest	Right	Healthy	18.64 (2.14)	17.45–19.82
		Left		19.64 (2.21)	18.42–20.87
	Contraction	Right		19.90 (2.29)	18.63–21.17
		Left		21.88 (2.4)	20.55–23.21
CSA (cm^2^)	Rest	Affected	Bulging	4.41 (0.8)	3.96–4.86
			Protrusion	3.73 (0.68)	3.35–4.12
			Extrusion	3.32 (0.86)	2.84–3.8
			Sequestration	2.94 (0.66)	2.58–3.31
	Contraction	Affected	Bulging	4.76 (1.07)	4.17–5.36
			Protrusion	4.39 (0.74)	3.98–4.8
			Extrusion	3.93 (0.85)	3.45–4.4
			Sequestration	3.39 (0.66)	3.03–3.76
AP (LM) Thickness (mm)	Rest	Affected	Bulging	18.51 (1.83)	17.49–19.53
			Protrusion	16.6 (2.01)	15.49–17.72
			Extrusion	15.45 (1.5)	14.61–16.28
			Sequestration	15.06 (1.33)	14.31–15.8
	Contraction	Affected	Bulging	19.55 (1.63)	18.64–20.45
			Protrusion	18.04 (1.77)	17.06–19.02
			Extrusion	16.04 (1.66)	15.12–16.96
			Sequestration	15.64 (1.4)	14.87–16.42

Abbreviation. CI: Confidence Interval; CSA: Cross-sectional area; AP: Anteroposterior; LM: Lumbar multifidus.

**Table 3 medicina-57-00699-t003:** LM muscles CSA and thickness changes comparison between different participant groups (at rest/contraction).

Muscles	Compared Patient Groups		Muscle’s State	Mean Difference	*p*-Value
CSA (cm^2^)	Healthy	Bulging	Rest	0.03	1
			Contraction	0.4	1
		Protrusion	Rest	0.71	0.12
			Contraction	0.77	0.11
		Extrusion	Rest	1.12	0.001
			Contraction	1.23	0.001
		Sequestration	Rest	1.5	0.001
			Contraction	1.77	0.001
AP(LM) (mm)	Healthy	Bulging	Rest	0.13	1
			Contraction	0.35	1
		Protrusion	Rest	2.04	0.02
			Contraction	1.86	0.05
		Extrusion	Rest	3.19	0.001
			Contraction	3.86	0.001
		Sequestration	Rest	3.58	0.001
			Contraction	4.26	0.001

**Table 4 medicina-57-00699-t004:** The relation between the LM muscles CSA and thickness with pain and disability in patient groups (at rest/contraction).

Muscles	Muscle’s State	Pain		Disability	
		*r*	*p*	*r*	*p*
CSA (cm^2^)	Rest	−0.49	0.001	−0.59	0.001
	Contraction	−0.44	0.001	−0.61	0.001
AP (LM) Thickness (mm)	Rest	−0.50	0.001	−0.68	0.001
	Contraction	−0.57	0.001	−0.68	0.001

Abbreviations. *r*: Pearson correlation coefficient; *p*: *p*-value.

## Data Availability

No new data were created in this study. Data sharing is not applicable to this article.

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
