# Peer review of "Lumbar Multifidus Muscle Morphology Changes in Patient with Different Degrees of Lumbar Disc Herniation: An Ultrasonographic Study"

_medicina, 2021, doi:10.3390/medicina57070699_

Round 1
Reviewer 1 Report
This work is interesting and concerns the importance and dimensions of the multifidus muscle depending on the degree of herniation of the intervertebral disc.
After revision, this article is clearer and more understandable.
Reviewer 2 Report
1. Small study-
2. Shows association of weak muscle mass more likely from disuse than from pain or disc disease.
3. Please include MRI definition of disc bulge, protrusion, sequestration.
4. reference was made to co-relation of muscle size to subjects disability but I see no such data collected about them.
5. tables and whole article can be more succinct, looks confusing to me.
Reviewer 3 Report
- Summary
This cross-sectional study investigated 45 patients with lumber disc herniation (LDH) and 15 healthy controls to measure cross-sectional area and thickness of lumbar multifidus muscles using ultrasonography. Authors identified the patients with LDH had smaller lumbar multifidus muscle at rest and during contraction compared to healthy controls. In addition, authors insisted the patients with severe LDH significantly associated with decreased muscle size.
- General Comments
In my view, the central topic of this study, “back muscle change of the patients with neurological symptoms” is an important and interesting subject to establish the adequate physical therapy for such patients. However, there are some comments:
- Authors may misunderstand the classification of LDH (bulging, protrusion, extrusion, and sequestration). The classification is just regarding to the anatomical characteristic of the LDH and does not indicate the size and severity of the LDH. This could critical flows of this study.
- The symptom duration and the severity of the symptoms could be major keys for the atrophy of the back muscle. I found there was the significant differences among the LDH group in the symptom duration that could role as a big confounding factor. On the other hand, the symptoms were not assessed in this study. Current results should be adjusted some factors including the symptom duration and the severity of the symptoms, otherwise the message from current study might not be useful for our clinical setting.
- I have a question about the reproducibility of the ultrasonography. Author must repot the inter- and intra-operator reliability. In addition, there was no information about the method to standardized the measurement.
III. Specific Comments
Please refer to this list of comments/suggestions to improve the quality of your manuscript.
- Abbreviations should be spelled out at the point of their first occurrence in the manuscript, after which only the abbreviation can be used throughout the rest of the document.
- There are several uncommon acronyms used throughout the document, which might confuse readers. The use of standard acronyms and abbreviations is recommended for clarity and easy readability.
Round 2
Reviewer 3 Report
I found that the authors have well addressed responses to the reviewer’s concerns and significantly improved the manuscript. However, authors just evaluated the univariate analysis which cannot exclude the effect of confounding factor. I still think that the severity of symptoms and duration of symptoms, but not the classification of LDH, may be associated with the CSA.Author Response
Please see the attachment.
